# TATKC: A Temporal Graph Neural Network for Fast Approximate Temporal Katz Centrality Ranking

## ABSTRACT

Numerous real-world networks are represented as temporal graphs, which capture the dynamics of connections over time. Identifying important nodes on temporal graphs has a plethora of real-life applications, such as information propagation and influential user identification, etc. Temporal Katz centrality, a popular temporal metric, gauges the importance of nodes by taking into account both the number of temporal walks and the timespan between the interactions. The computation of traditional temporal Katz centrality is computationally expensive, especially when applied to massive temporal graphs. Therefore, in this paper, we design a temporal graph neural network to approximate temporal Katz centrality computation. To the best of our knowledge, we are the first to address temporal Katz centrality computation purely from a learning-based perspective. We propose a time-injected self-attention model that consists of two phases. In the first phase, we utilize a time-injected self-attention mechanism to acquire node representations that encompass both structural information and temporal relevance. The second phase is structured as a multi-layer perceptron (MLP) which uses the learned node representation to predict node rankings. Furthermore, normalization and neighbor sampling strategies are integrated into the model to enhance its overall performance. Extensive experiments on real-world networks demonstrate the efficiency and accuracy of TATKC. Particularly, TATKC achieves an accuracy of up to 91.35% for the top-1% predictions and a prediction time of less than 200 seconds on a temporal graph with 4 million nodes and 22 million edges. Compared with the state-of-the-art exact TKC computation method, TATKC is capable of up to around 6 speedup.

## CCS CONCEPTS

• **Theory of computation** → **Graph algorithms analysis**; • **Computing methodologies** → *Neural networks.*

## KEYWORDS

Temporal Graph, Temporal Katz Centrality, Temporal Graph Neural Network, Self-attention

**ACM Reference Format:**
Anonymous Author(s). 2024. TATKC: A Temporal Graph Neural Network for Fast Approximate Temporal Katz Centrality Ranking. In *Proceedings of the ACM Web Conference 2024 (WWW '24)*. ACM, New York, NY, USA, 11 pages. https://doi.org/XXXXXXX.XXXXXXX

**Relevance to Web Research:** Temporal Katz centrality can be applied to web link graphs, social networks, etc. It enhances various aspects of web-related tasks, including search, recommendations, security, and understanding the dynamics of information flow.

## 1 INTRODUCTION

Many real-life networks are modeled as temporal graphs, such as web-based social networks, communication networks, transportation networks, etc., where interactions between nodes evolve over time. The use of centrality metrics to identify important nodes on the temporal graphs has a wide range of applications, such as rumor control [45], personalized content recommendation [28], identification of super-propagators in virus propagation [14], and identifying malicious network traffic [38]. The current temporal graph node centrality can be mainly divided into two categories, (i) optimal path-based temporal centrality, such as temporal betweenness centrality [9] and temporal closeness centrality [30]. (ii) Walk-based temporal centrality, e.g., temporal PageRank [35], temporal walk centrality [31], and temporal Katz centrality (TKC) [3]. Temporal PageRank is a generalization of PageRank for temporal networks. Temporal walk centrality counts random walks passing through a node. TKC is the weighted sum of temporal walks ending at a node, it quantifies the ability of a node to send out or receive information along the links. Considering that in real-life applications, messages do not always propagate along the optimal path, such as the spread of fake news in social networks and the spread of infectious diseases in contact networks [31], we focus on the problem of classic walk-based temporal centrality, TKC computation, which has widely influenced the study of social networks.

Traditional temporal Katz centrality computation involves traversing all edges of the entire temporal network and continuously updating node centrality values during the process. This computational approach results in significant runtime delays for TKC calculations on temporal graphs. More advanced data-centric techniques are needed to calculate TKC efficiently. Recently, much attention has been concentrated on how to learn graph algorithms by exploiting the family of Deep Graph Networks (DGNs), e.g., Bellman-Ford [44], graph clustering [26], which shows that it is indeed possible to train DGNs to execute classical graph algorithms. There exist models for betweenness centrality prediction on static graphs [16, 25]. However, it is important to note that these models, designed for static topology, do not possess the capability to accurately account for the impact of temporal fluctuations on node importance. We attempt to train neural networks to incorporate temporal node ranking reasoning by teaching them to execute classical TKC algorithms. To the best of our knowledge, no prior research has tackled TKC computation with a purely deep-learning model.

Designing a model for TKC is a non-trivial task due to the complex nature of temporal graphs and the computational challenges involved. Specifically, two challenges should be addressed.

***Challenge 1: How to capture complex temporal dependencies and effectively simulate the TKC computation process in the modeling?***

Existing temporal neural networks are mostly tailored for specific tasks other than TKC, hence they often lack the capability to effectively and efficiently simulate the TKC computation process. Essentially, TKC is a decay function influenced by both time elapsed and the length of temporal walks. To accurately simulate the temporal walks, assign weights to them, and accumulate the contribution from all weighted temporal walks, we employ a self-attention mechanism in the modeling process. To emulate the time decay process, we utilize continuous time encoding to capture the temporal relevance between edges. Based on these, we propose a temporal graph neural network model, called TATKC (Time-injected self-Attention model for TKC), which comprises two phases. In the first phase, we employ the time-injected self-attention mechanism to learn information-rich embedding vectors, which capture structural and temporal information in the original temporal graph. The second phase is structured as a multi-layer perceptron (MLP), which utilizes the acquired embedding vectors to predict node rankings based on TKC values. A pair-wise ranking loss function is employed to train the model. The model parameters undergo end-to-end training using various real-life temporal graphs, each annotated with ground truth TKC values for all nodes. Once trained, the resulting ranking model can be applied to any unseen temporal network for node importance assessment in terms of TKC values.

***Challenge 2: The computations involved in graph neural networks (GNNs) can be computationally intensive and time-consuming. How to efficiently train and predict TKC while preserving accuracy?***

Aggregating all neighbors in TATKC can be time-consuming, especially for massive graphs. Choosing the right aggregation method is crucial for both computational efficiency and model performance. we've observed that high TKC node values predominantly originate from a subset of crucial neighboring nodes. Consequently, we introduce a range of neighbor aggregation sampling methods, among which the degree-based neighbor sampling method proves highly effective in aggregating information from these influential neighboring nodes while excluding aggregation of neighbor nodes with little contribution. This reduces the complexity from $O(\mathcal{L} \cdot |E|)$ to $O(\mathcal{L} \cdot |V| \cdot \overline{|Sample|})$, where $|V|$ and $|E|$ are the number of nodes and temporal edges, respectively, $\mathcal{L}$ represents the number of model layers, and $\overline{|Sample|}$ indicates the average number of sampled neighbors. In our experiments, we found that TATKC can efficiently process massive graphs with tens of millions of edges, whereas TATKC without sampling encounters memory overflow issues and cannot run on such large datasets. Furthermore, practical testing on smaller datasets demonstrated that our sampling strategy significantly reduces graphics memory consumption by over 60%. Moreover, during the aggregation process, we have noticed that many nodes tend to aggregate neighbors that share strong similarities, leading to the over-smoothing phenomenon. To mitigate this issue, L2 normalization is applied to the aggregated node features. In brief, our contributions are summarized below.

- We propose an efficient and scalable temporal graph neural network model TATKC. Our model is inductive, capable of training on small graphs and subsequently making direct predictions of TKC rankings for massive graphs. To the best of our knowledge, we are the first work addressing TKC ranking from a purely learning-based approach.

- The proposed model TATKC is a two-phase temporal graph neural network. In the first phase, it employs a time-injected self-attention mechanism to learn node representations, mirroring the TKC calculation process. Additionally, to mitigate the over-smoothing phenomenon, L2 normalization is incorporated. In the second phase, the learned embedding vectors are utilized for node ranking score predictions. To improve computational efficiency, we introduce a range of neighbor sampling methods, among which the degree-based sampling proves highly effective in aggregating information from crucial neighboring nodes while excluding aggregation of neighbor nodes with little contribution.

- Extensive experiments are conducted to show the efficiency and effectiveness of the model. TATKC is trained on 50 small temporal graphs with less than 4 minutes, and tested on 8 real-life temporal networks in different scales, achieving an accuracy of up to 91.35% for the top-1% predictions and a prediction time of less than 200 seconds on a temporal graph with 4 million nodes and 22 million edges. Compared with the state-of-the-art exact method, TATKC is capable of up to around 6 speedups.

## 2 RELATED WORK

**Centrality computation on static graphs.** On static graphs, the commonly used centrality includes degree centrality [18], betweenness centrality [17], closeness centrality [2], PageRank [7] and Katz centrality [19], etc. A comprehensive survey is provided in [37]. Betweenness and closeness centralities are the shortest path-based metrics. For betweenness centrality, the representative literature is the Brandes algorithm [5] and its variants [1, 15, 36]. For closeness centrality, Saxena [37] offered a survey. PageRank [7] and Katz centrality [19] are walk-based metrics. PageRank ranks web pages by performing random walks. Katz centrality measures influence by taking into account the total number of walks between a pair of nodes. However, the computational costs of both shortest path-based and walk-based centralities become prohibitive for large-scale real-world networks. As a result, numerous approximation algorithms have been developed. Approximation algorithms can be categorized into traditional graph approximation algorithms [4, 6, 8, 10, 11, 13, 22–24, 27, 33, 34, 41] and deep learning-based approximation algorithms [16, 25]. Traditional approximation methods are usually based on sampling. Deep learning algorithms are mainly designed for betweenness centrality computation. DrBC [16] employs a recurrent neural network model, specifically the gated recurrent unit (GRU), to model static graphs. GNN-Bet [25] uses constrained message passing of node features to approximate betweenness centrality. Both DrBC and GNN-Bet exhibit limitations in scalability on large-scale graphs, as they encounter issues related to graphics memory insufficiency when dealing with massive graphs.

**Centrality computation on temporal graphs.** some pieces [3, 9, 20, 29–32, 35, 39] consider centrality on temporal graphs where edges exist at specific points in time. For temporal graphs, various optimal temporal paths (e.g., earliest arrival path, fastest path)

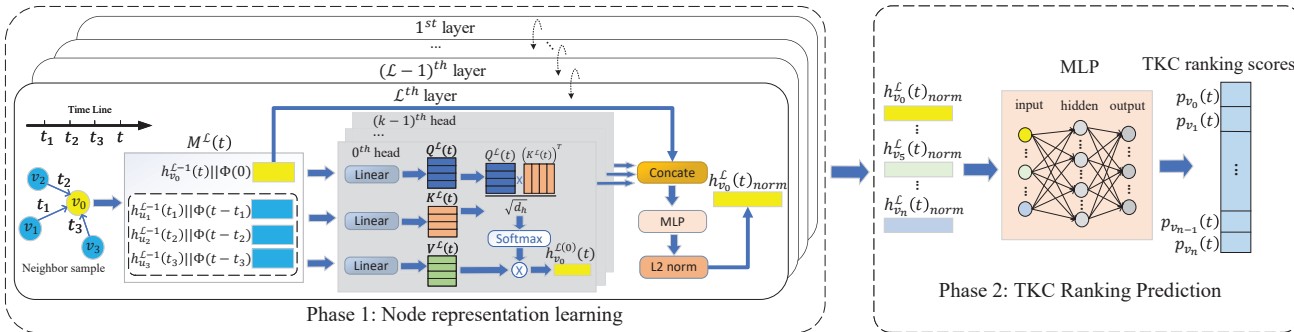

**Figure 1: Model architecture**

were explored in [42]. Then optimal path-based centrality such as temporal betweenness centrality [9] and temporal closeness centrality [30] were studied. Crescenzi et al. [12] proposed a sampling-based approximation algorithm for temporal closeness centrality computation. Oettershagen et al. [30] proposed TC-ALL for computing the exact temporal closeness centrality values. Tsalouchidou et al. [39] proposed a temporal path type that combined path length and temporal distance, and calculated betweenness centrality values on the static window with a fix-length. Buß et al. [9] introduced TGB, which first constructed the predecessor graph, and then extended Brandes' theory to the predecessor graph for calculation temporal betweenness centrality. In addition, some works adapt the walk-based centrality to temporal graphs. Hence temporal walk-based centralities, e.g., temporal PageRank [35], temporal walk centrality [31], and temporal Katz centrality [3, 32], are proposed. Oettershagen et al. [31] proposed temporal walk centrality, which counts temporal walks passing through a node. Ogura et al. [32] first generalized the Katz centrality to Markovian temporal networks. Béres et al. [3] presented temporal Katz centrality, which is the weighted sum of all temporal walks. Notably, there is a lack of current efforts to apply deep learning models to approximate walk-based centrality computation on temporal graphs.

## 3 PRELIMINARIES

*Definition 3.1.* (*Temporal Graph*). A directed temporal graph is defined as $G = (V, E, T)$, where $V$, $E$, and $T$ are the sets of the vertices, the temporal edges, and the timestamps, respectively. Specifically, a temporal edge $e = (u, v, t)$ represents an instantaneous event from $u$ to $v$ taking place at time $t \in T$.

*Definition 3.2.* (*Temporal Walk*). A temporal walk $z$ from node $u_0$ to $u_n$ is an ordered series of nodes and temporal edges in a temporal network, represented by $z = (u_0, u_1, t_1), (u_1, u_2, t_2), \ldots, (u_{n-1}, u_n, t_n)$, such that $\forall 1 \le i < n, t_i < t_{i+1}$. It is seen that temporal walks encapsulate the chronological interactions between nodes, highlighting their temporal attributes.

*Definition 3.3.* (*Temporal Katz Centrality* [3]). Given a temporal graph $G = (V, E, T)$, the temporal Katz centrality of a node $u \in V$ at time $t$ is the weighted sum of all temporal walks that end in node $u$, denoted by:

$$r_u(t) := \sum_{v} \sum_{temporal\ walks\ z\ from\ v\ to\ u} \Gamma(z, t) \qquad (1)$$

where $\Gamma(z, t)$ is the weight of temporal walk $z$ at time $t$. For a temporal walk $z = (u_0, u_1, t_1), (u_1, u_2, t_2), \ldots, (u_{n-1}, u_n, t_n)$, its weight $\Gamma(z, t)$ is represented as follows:

$$\Gamma(z, t) := \prod_{i=1}^{n} \varphi(t_{i+1} - t_i) \qquad (2)$$

Here, $\varphi$ is a weighting function, and when $i = n$, $t_{n+1} = t$.

According to the above definition, it is seen that the temporal Katz centrality formula depends on the selection of the weighting function $\varphi(\tau)$. The literature [3] proposes two important functions.

Function (i): $\varphi(\tau) = \beta$ is constant.

Function (ii): $\varphi(\tau) = \beta \cdot exp(-c\tau)$ is an exponential function, where $\beta < 1$ and $c < 1$ are constants.

If Function (i) is chosen, then the temporal Katz centrality is irrespective of time elapsed. In contrast, if Function (ii) is selected, then the temporal Katz centrality involves both a decay proportional to the length of the temporal walk and an exponential decay of time elapsed since the first interaction $t_1$ over the temporal walk occurred, which is capable of capturing the temporal decay of information spreading and propagation. Hence, in the paper, we use function (ii), i.e., $\varphi(\tau) = \beta \cdot exp(-c\tau)$, then $r_u(t)$ in Eq. 1 is rewritten as:

$$r_u(t) := \sum_{v} \sum_{temporal\ walks\ z\ from\ v\ to\ u} \prod_{i=1}^{n}(\beta \cdot exp(-c(t_{i+1} - t_i)))$$

$$:= \sum_{v} \sum_{temporal\ walks\ z\ from\ v\ to\ u} (\beta)^n \cdot exp(-c(t - t_1)) \qquad (3)$$

## 4 PROPOSED MODEL

As shown in Definition 3.3, TKC calculation involves summarizing all temporal walks with weights. Therefore, the corresponding learning framework needs to iteratively aggregate neighbor information, and different neighbors may have varying contributions to the target node. Hence we decide to use the self-attention mechanism [40] to learn the different weights of neighbors and aggregate them. In addition, it is noticed that the weights of temporal walks are influenced by a decay function that considers not only walk length but also time information, hence the proposed model should be adept at modeling and integrating the most critical temporal aspects of the temporal graph. Based on this, we introduce our

model TATKC, which consists of two phases, as shown in Figure 1. The first phase leverages the self-attention mechanism fused with time encoding to learn node representations. The second phase utilizes the learned node representations to predict node rankings based on TKC values with MLP. Next, we introduce each module of TATKC in detail.

## 4.1 Node Representation Learning

Self-attention mechanism assigns different weights to neighboring nodes when aggregating node features. Since it captures sequential information only through positional encoding, temporal features cannot be handled, we replace positional encoding with time encoding. Hence, in the first phase, we employ the time-injected self-attention to learn embedding vectors, which capture both structural and temporal information in the original temporal graphs.

*4.1.1 Continuous time encoding.* We employed the approach introduced in [43], which applies Bochner's theorem and Monte Carlo integral to map a temporal domain $T = [0, t_{max}]$ (where $t_{max}$ is the maximum timestamp among all the temporal edges in $G$) to the $d$-dimensional vector space. Specifically, let $\Phi(t)$ be the time encoding of any time point $t \in T$. Given two time points $t_1, t_2 \in T$, a temporal kernel function $\mathcal{K}(t_1, t_2)$ which measures the timespan between $t_1$ and $t_2$, can be expressed by the inner product of their time encodings,

$$\mathcal{K}(t_1, t_2) = \psi(t_1 - t_2) = \Phi(t_1) \cdot \Phi(t_2) \approx \Phi_{d_T}(t_1) \cdot \Phi_{d_T}(t_2) \quad (4)$$

and

$$\Phi_{d_T}(t) = \sqrt{\frac{1}{d_T}} \left[ \cos(\omega_1 t), \sin(\omega_1 t), ..., \cos(\omega_{d_T} t), \sin(\omega_{d_T} t) \right]^\top \quad (5)$$

where $d_T$ is the finite dimension and $\boldsymbol{\omega} = (\omega_1, \ldots, \omega_{d_T})^\top$ are learnable parameters.

*4.1.2 Time-injected self-attention.* Let $h_v \in \mathbb{R}^d$ ($d$ is the dimension of features) be the structural feature of node $v$. Traditional self-attention uses positional encoding, here, time-injected self-attention utilizes time encoding $\Phi(t)$ to replace the positional encoding. Given a temporal graph $G = (V, E, T)$, a node $v \in V$ is split into a set of instances, denoted by $S(v) = \{(v, t) \mid t \in T_v\}$, where $T_v$ is the set of distinct time instances attached with the outgoing edges of $v$. We define the representation of $(v, t)$ as $h_v(t) = h_v || \Phi(t)$, where $||$ denotes concatenation operation. To emulate the process of aggregating temporal walks in TKC calculation, for a node $v$ at time $t$, the in-neighbors from which an interaction occurred before time $t$ are aggregated. Let $N_{in}(v)_{<t} = \{(u, t_u) | (u, v, t_u) \in E \land (t_u < t)\}$ be the set of such in-neighbors of $v$. Hence the input matrix of $l^{th}$ attention layer is:

$$M^l(t) = [h_v^{l-1}(t) || \Phi(0), h_{u_1}^{l-1}(t_1) || \Phi(t-t_1), ..., h_{u_n}^{l-1}(t_n) || \Phi(t-t_n)]^T \quad (6)$$

where $\forall 1 \leq i \leq n$, $(u_i, t_i) \in N_{in}(v)_{<t}$, and $h_v^{l-1}(t)$ is the hidden representation output for $(v, t)$ (i.e., node $v$ at time $t$) from the $(l-1)^{th}$ layer. Note that Eq. 6 uses the encoding of the time difference because Eq. 3 is a function of the time difference.

Then $M^l(t)$ are fed into three linear projections to obtain "query" matrix $Q^l(t) = [M^l(t)]_0 \cdot W_Q$, "key" matrix $K^l(t) = [M^l(t)]_{1:n} \cdot W_K$, and "value" matrix $V^l(t) = [M^l(t)]_{1:n} \cdot W_V$, where $W_Q, W_K, W_V \in$

$\mathbb{R}^{(d+d_T) \times d_h}$ are projection matrices capturing interactions between time encoding and node features, and $d_h$ is the dimension of hidden layer. Next, the scaled dot-product attention is used in self-attention layers, i.e.,

$$\text{Attn}(Q^l(t), K^l(t), V^l(t)) = softmax\left(\frac{Q^l(t)(K^l(t))^T}{\sqrt{d_h}}\right)V^l(t) \in \mathbb{R}^{d_h} \quad (7)$$

To improve training performance, multi-head attention is employed. In the sequel, to get $h_v^l(t)$, the results of multi-head attention and the node representations learned in the $(l-1)^{th}$ layer are aggregated using a two-layer MLP with ReLU activation function.

$$h_v^l(t) = \text{MLP}(h_v^{l(0)}(t) || h_v^{l(1)}(t) || ... || h_v^{l(k-1)}(t) || h_v^{l-1}(t)) \quad (8)$$

where $k$ is the number of heads, and $h_v^{l(j)}(t) (0 \leq j < k)$ is the dot-product self-attention output of $j^{th}$ head, computed by Eq. 7.

A single attention layer described above aggregates the localized one-hop neighborhood, and $\mathcal{L}$ hops neighborhood and time encoder information are aggregated by stacking $\mathcal{L}$ such attention layers.

Note that as defined in Definition 3, TKC collects temporal walks, i.e., walks that respect time order. Hence for the node $v$ at time $t$, the self-attention mechanism aggregates neighbors that satisfy time constraint, i.e., interaction occurs before $t$. Then a natural question is raised, how to set $t$? There are two cases.

case (i): If $v$ is the target node, then $t = t_{max}$, where $t_{max}$ is the maximum timestamp among all the temporal edges. Thus the model can predict the current TKC values.

case (ii): Otherwise $v$ is the intermediate node, then $t = min\{t | (v, w, t) \in E_{out}(v)\}$, where $E_{out}(v)$ refers to the set of $v'$s outgoing temporal edges. In this way, temporal walks are aggregated, and walks that are unordered are excluded.

**Normalization.** The node representations obtained by the above-mentioned self-attention mechanism demonstrate unsatisfactory performance, especially in predicting the top-1% key nodes. This issue is potentially due to the over-smoothing phenomenon, where nodes within the graph progressively become more similar, rendering them virtually indistinguishable. The underlying problem stems from the aggregation of neighbor information. Many nodes tend to aggregate neighbors that share strong similarities, leading to closely clustered representations. To address this issue, we implement L2 normalization on the aggregated node features.

$$h_v^l(t)_{\text{norm}} = \frac{h_v^l(t)}{\|h_v^l(t)\|_2} = \frac{h_v^l(t)}{\sqrt{x_1^2 + x_2^2 + \ldots + x_n^2}} \quad (9)$$

where $h_v(t) = (x_1, x_2, \cdots, x_n)$. L2 normalization ensures that features have consistent scales and prevents certain features from dominating others, ultimately improving the ability to differentiate among the top-N% TKC nodes, especially in top-1%.

*4.1.3 Strategy of neighbor sampling.* For temporal graphs with a power-law degree distribution, the nodes with the maximum degree, often referred to as "hubs," always dominate the computation costs. This is because these hub nodes have an exceptionally high number of connections (the neighbor degrees of these hubs can tend to be nearly the size of the entire graph), which can lead to time-consuming aggregation. To prevent the number of neighbors from increasing without limitation as time flows, TATKC samples

neighbors for information aggregation to improve efficiency and stabilization. We consider five different neighbor sampling methods.

(i) **Uniform sampling**: Neighbors are selected with equal probability. It's straightforward to sample neighbors. However, this uniform sampling approach can lead to information overlap, where the same neighboring nodes are chosen multiple times, potentially leading to bias in the learned representations. This can cause the model to overlook the importance of other neighbors.

(ii) **Recent interaction sampling**: Neighbors are chosen based on the recency of their interactions. Specifically, neighbors that have the most recent interactions with the focal node are chosen preferentially. It enables capturing the most up-to-date interactions. However, this approach may not adequately capture long-term trends that are important for comprehending network evolution.

(iii) **Farthest interaction sampling**: Unlike recent interaction sampling, which prioritizes recent interactions, farthest interaction sampling selects neighbors that have the farthest interaction time, which gains insight into learning how the interactions of neighbors change over time. However, information from neighbors that are too distant may have become outdated.

(iv) **Expanded neighbor sampling**: It considers the temporal order of interactions between the target node and its in-neighbors. In-neighbor instances are sorted along the time axis. The sampled neighbors are not adjacent in time but are separated by "$r$ - 1" other neighbors in chronological order, where $r = \frac{number\ of\ instances}{number\ of\ samples}$. It allows the model to access neighbor information from various time points. However, if there are substantial differences among neighbors from different time points, this approach may introduce potential sources of noise.

(v) **Degree-based sampling**: It chooses to sample a user-specified number of nodes with the highest degrees. Since nodes with high TKC values are often associated with a high degree, the degree-based sampling method is useful for targeting highly connected nodes with high TKC values.

Based on these methods, the pseudo-code of the first phase is illustrated in Algorithm 1 in Appendix A.

## 4.2 TKC Ranking Prediction

In the second phase, we take the node representations generated during the first phase as inputs for forecasting the TKC ranking scores of nodes, denoted as $p_v(t)$. The process is facilitated through the utilization of a three-layered multi-layer perceptron (MLP) integrated with rectified linear unit (ReLU) activation functions:

$$p_v(t) = MLP(h_v^{\mathcal{L}}(t)_{\text{norm}}), v \in V \qquad (10)$$

**Loss function.** As our task can be viewed as a ranking task, we utilize the pair-wise ranking loss function [25] to train the model for approximating TKC ranking scores of nodes. This loss function is applied to compute the dissimilarity between the predicted ranking scores and ground truth TKC values. Given an arbitrary node pair $(v_1, v_2)$, $v_1, v_2 \in V$, our model output scores $p_{v_1}(t_{max})$ and $p_{v_2}(t_{max})$ and we abbreviate it as $p_1$ and $p_2$, respectively. Let $y_1$ and $y_2$ be the corresponding ground truth TKC values of $v_1$ and $v_2$. As such, the pair-wise ranking loss function is formulated as follows:

$$\text{Loss}(p_1, p_2, y) = \max(0, -y \cdot (p_1 - p_2) + \text{Margin}), \qquad (11)$$

### Table 1: Statistics of the datasets

| Dataset | $|V|$ | $|E|$ | $|T|$ |
|---|---|---|---|
| tgwiktionary | 33,968 | 81,516 | 67,065 |
| mlwikiquote | 43,889 | 142,340 | 137,389 |
| mgwikipedia | 220,064 | 750,811 | 736,680 |
| plwikiquote | 581,646 | 1,472,273 | 1,452,278 |
| ltwiktionary | 689,678 | 1,693,277 | 1,633,334 |
| zhwiktionary | 1,347,094 | 5,276,371 | 4,448,306 |
| warwikipedia | 2,877,072 | 6,145,080 | 5,918,117 |
| mgwiktionary | 4,064,239 | 22,720,139 | 19,759,219 |

$$y = \begin{cases} 1 & \text{if } y_1 > y_2; \\ 0 & \text{if } y_1 = y_2; \\ -1 & \text{if } y_1 < y_2; \end{cases}$$

where Margin refers to a difference value, which is a hyper-parameter. A larger value for this parameter implies a greater expected separation between $v_1$ and $v_2$.

## 4.3 Complexity Analysis

**Training complexity.** The time complexity scales linearly with the number of training iterations, making it difficult to provide a precise theoretical analysis. However, in empirical experiments, TATKC demonstrates rapid convergence, as illustrated in Table 3.

**Inference complexity.** It is primarily dominated by the first phase, which costs $O(\mathcal{L} \cdot |V| \cdot \overline{|Sample|})$. $\mathcal{L}$ is the number of model layers, $|V|$ and $|E|$ are the number of nodes and temporal edges, respectively, and $\overline{|Sample|}$ indicates the average number of sampled neighbors. On massive real-world graphs, $|V| \cdot \overline{|Sample|} << |E|$.

**Space complexity.** The space complexity of TATKC is characterized by the graphic memory requirements needed to store node features during both training and inference. To control graphic memory consumption, we adopted a similar approach to batch sub-graph processing as described in [43]. In summary, the spatial complexity of our model is $O(b \cdot (\overline{|Sample|})^{\mathcal{L}} \cdot d)$, where $b$ is the batch size of the sub-graph.

## 5 EXPERIMENTS

### 5.1 Experimental Setup

*5.1.1 Datasets.* We utilize 8 real datasets from Konect[1] for evaluation. The statistics are summarized in Table 1. The wiktionary, wikiquote, and wikipedia datasets represent edit networks specific to their respective platforms. These datasets encompass both editors and pages, interconnected through edit events or editor communication. Within these networks, edges can be classified into two categories: a majority of edges between users and pages are accompanied by timestamps that precisely indicate when edits occurred, while a small subset of edges connecting editors is associated with timestamps that denote when the editors initiated communication. Our training dataset consists of 50 real-world datasets from Konect, each ranging from 1,000 to 12,000 nodes.

*5.1.2 Comparisons.* To the best of our knowledge, there are currently no TKC approximation algorithms or deep learning models.

---

[1]Konect is available at http://konect.cc/

Table 2: Top-N% accuracy($\times$ 0.01) and Kendall's tau (KT) scores($\times$ 0.01) on varying scale real-world temporal graphs. The bold results indicate the best performance. For each dataset, we report the mean and standard deviation over 5 tests.

| Dataset | Top-1% | | | Top-5% | | | Top-10% | | | Top-20% | | | KT | | |
|---|---|---|---|---|---|---|---|---|---|---|---|---|---|---|---|
| | TATKC | TATKC* | TGAT | TATKC | TATKC* | TGAT | TATKC | TATKC* | TGAT | TATKC | TATKC* | TGAT | TATKC | TATKC* | TGAT |
| tgwiktionary | **90.68±0.96** | 75.39±1.98 | 64.87±1.90 | **93.77±0.17** | 85.44±0.59 | 80.44±1.19 | **88.41±0.54** | 84.12±0.31 | 81.07±0.65 | **92.53±0.21** | 88.42±0.40 | 86.07±0.36 | **80.02±0.42** | 78.87±0.21 | 77.67±0.76 |
| mlwikiquote | **85.27±0.88** | 60.86±1.43 | 61.19±1.67 | **89.37±0.82** | 73.56±1.72 | 69.02±1.23 | **89.12±0.61** | 76.35±1.17 | 73.73±0.70 | **91.82±0.29** | 81.85±0.26 | 80.49±0.42 | **86.60±0.23** | 82.70±0.37 | 82.04±0.38 |
| mgwikipedia | **80.01±0.46** | 65.96±0.51 | 49.41±0.48 | **86.67±0.14** | 76.88±1.02 | 67.67±0.33 | **97.47±0.05** | 94.92±0.19 | 91.29±0.14 | **94.85±0.03** | 94.55±0.08 | 94.12±0.08 | **76.98±0.08** | 76.58±0.12 | 76.29±0.14 |
| plwikiquote | **85.34±0.18** | 73.33±0.93 | 66.98±0.58 | **85.08±0.02** | 74.37±0.38 | 70.95±0.24 | **84.76±0.27** | 76.02±0.18 | 73.69±0.23 | **85.97±0.18** | 78.74±0.22 | 77.47±0.12 | **82.49±0.17** | 79.60±0.03 | 79.03±0.04 |
| ltwiktionary | **88.14±0.82** | 60.45±0.13 | 56.48±0.21 | **94.32±0.08** | 89.51±0.08 | 83.99±0.13 | **95.29±0.05** | 94.03±0.04 | 92.67±0.11 | **94.98±0.03** | 94.48±0.05 | 94.38±0.02 | **70.68±0.04** | 70.34±0.06 | 70.17±0.04 |
| zhwiktionary | **72.20±0.41** | 57.48±0.11 | 55.40±0.18 | **91.48±0.31** | 82.31±0.68 | 73.42±0.31 | **88.89±0.36** | 83.13±0.29 | 79.13±0.31 | **91.88±0.03** | 89.93±0.14 | 87.90±0.17 | **84.90±0.03** | 83.13±0.29 | 82.14±0.11 |
| warwikipedia | **91.35±0.08** | 72.02±0.28 | 58.46±0.21 | **95.74±0.03** | 94.01±0.05 | 93.48±0.14 | **76.32±0.11** | 75.08±0.27 | 75.63±0.09 | **78.13±0.03** | 78.04±0.04 | 78.07±0.09 | **71.28±0.02** | 71.13±0.04 | 71.10±0.02 |
| mgwiktionary | **90.25±0.21** | 60.33±0.87 | 46.59±1.43 | **89.81±0.26** | 76.84±0.40 | 62.69±0.98 | **90.10±0.14** | 79.21±0.19 | 69.29±0.73 | **97.61±0.16** | 84.41±0.09 | 78.17±0.48 | **84.65±0.06** | 79.42±0.10 | 75.12±0.33 |

Hence we compare TATKC with the up-to-date exact TKC methods [3], a temporal node representation learning model TGAT [43], and two deep models [16, 25] designed for betweenness centrality rankings, which are detailed below.

- **ETKC**[2]: ETKC computes exact TKC values by Eq. 3, which serves as the ground truth. Follow [3], we set $\beta = 0.01$, $n = 2$. In Eq. 3, $c$ is a weight associated with the time span of the temporal walks, we set $c = \frac{1}{2(t_{max} - t_{min})}$, where $t_{max}$ and $t_{min}$ are the maximum and minimum timestamps among all the temporal edges in a graph, respectively.
- **DrBC**[3]: DrBC is an encoder-decoder based framework network. The encoder leverages the GRU network to represent each node as an embedding vector. The decoder converts the embedding vectors into scalars, signifying the node's relative ranking concerning its betweenness centrality values.
- **GNN-Bet**[4]: GNN-Bet is a graph neural network-based framework, which restricts the flow of feature information to the edges located on the shortest paths and uses constrained message passing of node features to approximate betweenness centrality.
- **TATKC***: TATKC* removes the L2 normalization setting in TATKC, while keeping all other configurations unchanged.
- **TGAT**[5]: TGAT is a temporal graph attention network designed for learning temporal node representations using time features and uniform neighbor sampling.

*5.1.3 Evaluation metrics.* We evaluate the model's effectiveness in terms of top-N% accuracy and Kendall's tau correlation [21].

Top-N% accuracy is defined as the percentage of overlap between the top-N% nodes as returned by an approximation method and the top-N% nodes considered as the ground truth:

$$\text{Top-N\%} = \frac{|\text{predicted top-N\% nodes} \cap \text{true top-N\% nodes}|}{\lceil |V| \times \text{N\%} \rceil} \quad (12)$$

where $|V|$ is the number of nodes, and $\lceil . \rceil$ is the ceiling function. In our paper, we mainly compare top-1%, top-5%, top-10%, and top-20%, as real applications often focus on highly influential entities.

Kendall's tau (KT) correlation is defined as (#concordant pairs - #discordant pairs) $/|V|(|V|-1)$, where #concordant pairs is the number of concordant pairs of centrality rankings; #discordant pairs is the number of discordant pairs of centrality rankings. KT scores range between -1 and +1, where 0 indicates that there is no

correlation between the two rankings; 1 indicates that there is a completely positive correlation between the two rankings; and -1 indicates that there is a completely negative correlation between the two rankings.

*5.1.4 Hyper-parameters.* The model is implemented in PyTorch, utilizing the Adam optimizer with a learning rate of 0.01. We configured the node and time embedding dimension as 128, set the number of neighbor samples to 20, and employed a model with 2 layers, conducting training for 15 epochs. To compute the loss during training, we followed the same configuration as in GNN-Bet, which involved randomly sampling node pairs equivalent to 20 times the total number of nodes in the graph.

*5.1.5 Hardware and software setup.* All experiments were conducted on a server equipped with an Intel Core i9-9900X CPU, 128 GB of RAM, and an NVIDIA GeForce RTX 2080Ti GPU with 11GB of graphics memory.

Next, the conducted experiments answer the following questions:

- $Q_1$. **Accuracy:** How is the accuracy of TATKC compared to the exact TKC rankings?
- $Q_2$. **Efficiency:** Is TATKC efficient both in the training and in predicting TKC rankings? Does it scale to massive networks?
- $Q_3$. **Factors Affecting Model Performance:** How do different factors (e.g., neighbor sampling method, the number of neighbors, etc.) affect the performance of our model?
- $Q_4$. **Node Ranking Correlation:** What is the level of correlation between the TKC rankings predicted by our model and betweenness centrality rankings predicted by other deep learning models?

## 5.2 Experimental Result

$Q_1$. **Accuracy.** We evaluate the accuracy of TATKC, TATKC* and TGAT using both top-N% and KT scores. The result is shown in Table 2. It is observed that TATKC outperforms TATKC* and TGAT across all datasets in terms of the top-N% accuracy, as well as KT scores. Particularly, in the case of top-1%, TATKC consistently outperforms both TATKC* and TGAT, with improvements ranging from 12% to 30% compared to TATKC* and 17% to 43% compared to TGAT. This observation highlights that incorporating L2 normalization effectively improves the model's capability to predict top-N% nodes, particularly in the top-1%. The prediction performance of TGAT is subpar, primarily because it struggles to sample suitable temporal neighbors that can adequately simulate the TKC computation process.

---

[2]Code of ETKC is available at https://github.com/ferencberes/online-centrality
[3]Code of DrBC is available at https://github.com/FFrankyy/DrBC
[4]Code of GNN-Bet is available at https://github.com/sunilkmaurya/GNN-Bet
[5]Code of TGAT is available at https://github.com/StatsDLMathsRecomSys/Inductive-representation-learning-on-temporal-graphs

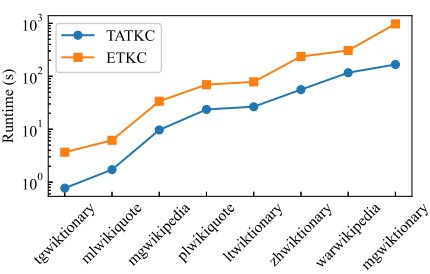

**Figure 2: Model efficiency**

**Table 3: Training time comparison**

| Model name | TATKC | DrBC | GNN-Bet |
|---|---|---|---|
| Training time (s) | 164.92 | 73071.55 | 1088.63 |

**Table 4: Model comparison(×0.01) on mlwikiquote (small-scale dataset)**

| Model name | Top-1% | Top-5% | Top-10% | KT |
|---|---|---|---|---|
| TATKC | **85.27** | **89.37** | 89.12 | 86.60 |
| DrBC | 76.94 | 51.67 | **100.0** | 31.60 |
| GNN-Bet | 41.54 | 80.08 | 65.54 | **97.02** |

**Table 5: Model comparison(×0.01) on ltwiktionary (middle-scale dataset)**

| Model name | Top-1% | Top-5% | Top-10% | KT |
|---|---|---|---|---|
| TATKC | 88.14 | 94.32 | 95.29 | 70.68 |
| DrBC | **100.0** | **100.0** | **100.0** | 6.48 |
| GNN-Bet | 42.68 | 77.27 | 85.56 | **98.47** |

$Q_2$. **Efficiency.** We evaluated the prediction time of TATKC against the exact TKC computation method ETKC. Figure 2 depicts the result. It is observed that TATKC is from 3 to 6 times faster than ETKC over different datasets. This is because, ETKC requires each node to traverse all temporal walks that reach it, continuously calculating temporal walk weights and iteratively updating the node's TKC value. In contrast, TATKC samples a subset of the direct neighbors for each node, aggregating information from a portion of its neighbors, thus reducing the time complexity. Particularly, TATKC completes TKC rankings of all nodes within 168 seconds on mgwiktionary dataset with 4 million nodes and 22 million edges.

Additionally, we investigated the training efficiency of TATKC. As there is currently no deep learning model available for calculating TKC, we compare TATKC with DrBC and GNN-Bet, which are betweenness centrality ranking prediction models. Since DrBC and GNN-Bet predict betweenness centrality for static graphs rather than temporal graphs, we omit the timestamps in the datasets during the calculation. According to [16] and [25], DrBC is trained on 1,000 synthetic graphs with node sizes in the range of 4000~5000;

**Table 6: Model comparison(×0.01) on mgwiktionary (massive-scale dataset): NA denotes memory overflow**

| Model name | Top-1% | Top-5% | Top-10% | KT |
|---|---|---|---|---|
| TATKC | **90.25** | **89.81** | **90.10** | **84.65** |
| DrBC | NA | NA | NA | NA |
| GNN-Bet | NA | NA | NA | NA |

GNN-Bet is trained on 400 synthetic graphs with 10,000 nodes. TATKC is trained on 50 real graphs with node sizes spanning 1000~12,000. Table 3 shows the training time of these models. We also report the top-1%, 5%, 10% accuracy, and KT scores across small, middle, and massive scale datasets, the results are illustrated in Tables 4, 5 and 6, respectively. The first observation is that, as shown in Table 3, TATKC can be trained in less than 4 minutes, which is 6 times faster than GNN-Bet and two orders of magnitude faster than DrBC. The reason is that, our model's training datasets are relatively smaller, and the neighbor sampling strategy is adopted to reduce computational costs, resulting in shorter training time.

The second observation is that, as seen in Tables 4 and 5, for small and middle scale datasets, DrBC achieves the highest top-N% accuracy on most cases, albeit with the trade-off of exhibiting the lowest KT scores. This is because, DrBC is designed to identify highly influential nodes with high betweeness centrality scores, disregarding the rankings among nodes with lower scores. Conversely, GNN-Bet attains the highest KT scores while lagging behind in terms of top-N% accuracy. The reason is that, GNN-Bet focuses on the betweenness centrality computation, which is based on shortest paths. To simulate information propagation along shortest paths, GNN-Bet selectively omits the aggregation of vertices with betweenness centrality scores of 0. While the proportion of these vertices is very large, overlooking them can lead to the model not fully understanding the graph topology, resulting in low accuracy. TATKC consistently demonstrates impressive and stable performance in both top-N% accuracy and KT scores. This is mainly because the time-injected self-attention mechanism effectively simulates the process of aggregating temporal walks in TKC computation defined in Eq. 3. As depicted in Table 6, for the massive-scale dataset, both DrBC and GNN-Bet encounter graphic memory overflow issues as they need to load the sparse adjacency matrix onto the GPU, which strains the available memory resources. In contrast, TATKC continues to perform efficiently and effectively, attributed to the utilization of the neighbor sampling strategy, allowing it to selectively aggregate crucial neighbor information, thus alleviating the memory burden on the GPU.

$Q_3$. **Factors Affecting Model Performance.** We investigated the effects of different neighbor sampling strategies on the model's prediction accuracy of top-1% and top-5% nodes. Tables 7 and 9 (Table 9 is shown in Appendix B) show the results. First, it is observed that uniform sampling performs the worst in all datasets. Degree-based sampling performs the best in 6 out of the 8 datasets in terms of top-1% accuracy, and it achieves the best top-5% accuracy in four datasets. On other datasets, top-1% or top-5% accuracy is close to the best result. The reason is that nodes with high degrees often dominate the TKC scores. These nodes are exactly obtained

**Table 7: Top-1% Accuracy(×0.01) under different sampling strategies**

| Dataset | Uniform | Recent Interaction | Farthest Interaction | Expanded Neighbor | Degree-based |
|---|---|---|---|---|---|
| tgwiktionary | 68.36 | 90.42 | 90.30 | 90.66 | **90.68** |
| mlwikiquote | 58.21 | **86.56** | 82.83 | 83.08 | 85.27 |
| mgwikipedia | 47.72 | 75.95 | 71.73 | 73.93 | **80.01** |
| plwikiquote | 66.16 | 84.83 | 85.11 | 84.74 | **85.34** |
| ltwiktionary | 54.71 | 88.89 | 89.03 | **89.08** | 88.14 |
| zhwiktionary | 54.07 | 61.65 | 68.45 | 65.85 | **72.20** |
| warwikipedia | 56.74 | 91.21 | 91.25 | 91.08 | **91.35** |
| mgwiktionary | 45.15 | 90.21 | 90.16 | 90.20 | **90.25** |

(a) tgwiktionary          (b) plwikiquote          (c) warwikipedia          (d) mgwiktionary

**Figure 3: Top-1% accuracy and prediction time vs. the number of samples**

**Table 8: KT correlation between the rankings of top-1% key nodes computed by TATKC and DrBC, TATKC and GNN-Bet**

| Dataset | DrBC vs TATKC | GNN-Bet vs TATKC |
|---|---|---|
| tgwiktionary | 0.0711 | 0.0137 |
| mlwikiquote | 0.0149 | 0.0113 |
| mgwikipedia | 0.1779 | 0.2305 |
| plwikiquote | -0.1138 | -0.1292 |
| ltwiktionary | -0.0017 | 0.0071 |

by degree-based sampling method and aggregated by self-attention, leading to high top-N% accuracy. As a result, we use the degree-based sampling strategy by default.

In addition, we also investigate the impact of the number of samples on the top-1% accuracy and prediction time. The number of samples is varied from 5 to 30, and Figure 3 illustrates the results on 4 datasets. First, it is observed that the prediction time increases with the growth of the number of samples. This is because more node information is involved in the aggregation. Additionally, we observe that top-1% accuracy first significantly increases and then gradually stabilizes as the number of samples ascends. This trend can be attributed to increased valid information with more samples. However, when the number of samples reaches a certain number, key neighborhood information has been all aggregated and the top-1% accuracy reaches the highest value. In the sequel, increasing the number of samples again does not improve the accuracy, but only increases the computation costs. There is a trade-off between accuracy and efficiency. As depicted in Figure 3, we set the number of samples to 20 by default to achieve high accuracy.

$Q_4$. **Node Ranking Correlation.** Table 8 shows the KT scores between the rankings of top-1% nodes computed by TATKC and DrBC, as well as TATKC and GNN-Bet. As observed, KT scores are between -0.1295 and 0.2305, which shows that rankings computed by TATKC and DrBC (or GNN-Bet) are pairwise dissimilar. This is because, DrBC and GNN-Bet ignore the timestamps on edges, and they measure the impact of shortest paths on node importance, while TKC considers the impact of all temporal walks.

Furthermore, we also conducted a case study to show the difference between TKC and betweenness centrality, and performed ablation experiments to dissect various components of TATKC. Due to space limitations, the case study and ablation experiments are provided in Appendix C and Appendix D, respectively.

## 6 CONCLUSION

In this paper, we explore the potential of temporal graph neural networks for approximating temporal Katz centrality rankings of nodes. To the best of our knowledge, we are the first to address TKC rankings from a purely learning-based approach. A two-phase model, named TATKC, is designed. In the first phase, TATKC uses continuous time encoding and time-injected self-attention to emulate the process of exact TKC computation and learn node representations. In the second phase, based on the learned node representations, TATKC predicts TKC ranking scores with MLP. To improve performance, normalization and neighbor sampling strategies are integrated. Extensive experiments on real-world networks demonstrate the efficiency and accuracy of TATKC. In the future, we plan to design models for approximating other complex temporal centrality measures, such as temporal closeness centrality or temporal betweenness centrality.

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

**Table 9: Top-5% Accuracy(×0.01) under different sampling strategies**

| Dataset | Uniform | Recent Interaction | Farthest Interaction | Expanded Neighbor | Degree-based |
|---------|---------|--------------------|--------------------|--------------------|--------------|
| tgwiktionary | 78.63 | 93.60 | 93.89 | **93.94** | 93.77 |
| mlwikiquote | 68.66 | **89.47** | 88.77 | 88.82 | 89.37 |
| mgwikipedia | 66.28 | 85.97 | 86.18 | 86.28 | **86.67** |
| plwikiquote | 70.27 | 85.03 | **85.11** | 85.07 | 85.08 |
| ltwiktionary | 83.31 | **94.36** | 94.32 | 94.27 | 94.32 |
| zhwiktionary | 73.62 | 91.43 | 91.41 | 91.13 | **91.48** |
| warwikipedia | 93.39 | 95.73 | 95.72 | 95.67 | **95.74** |
| mgwiktionary | 60.60 | 89.72 | 89.61 | 89.77 | **89.81** |

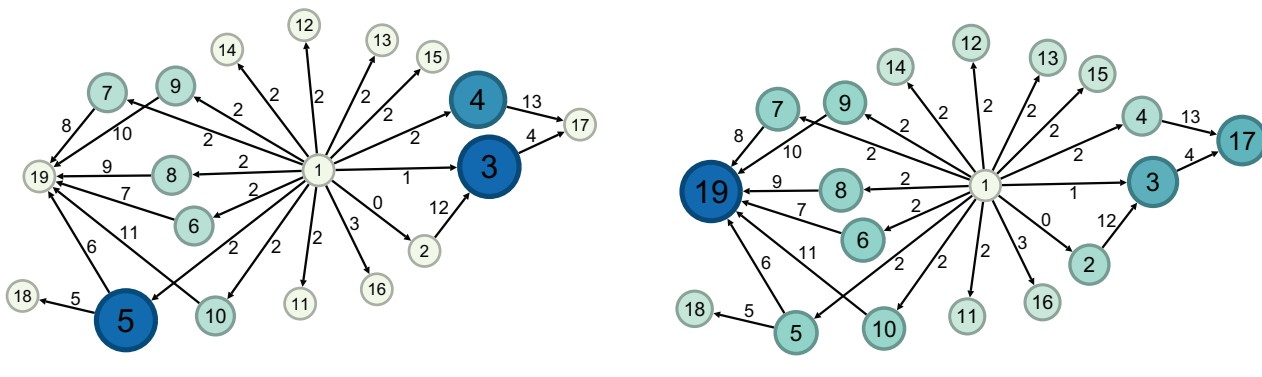

(a) Nodes ranked by BC                         (b) Nodes ranked by TKC

**Figure 4: Nodes ranked by BC and TKC on the subgraph of tgwiktionary**

## A  LEARNING NODE REPRESENTATION

Algorithm 1 outlines the entire process of TATKC's first-stage node representation learning.

---

**Algorithm 1:** Learning Node Representation

---

**Input:** Network $G = (V, E, T)$; model layer $\mathcal{L}$; the number
      of heads $k$, time $t \in \mathbb{N}$.
**Output:** Node representations $h_v^{\mathcal{L}}(t)_{norm}$.

1  Initialize $h_v^0(t)$;
2  **for** *layer $l = 1$ to $\mathcal{L}$* **do**
3     **for** $v \in V$ **do**
4        Neighbor sampling and form matrix $M^l(t)$ by Eq. 6;
5        $Q^l(t) = [M^l(t)]_0 \cdot W_Q$; $K^l(t) = [M^l(t)]_{1:n} \cdot W_K$;
        $V^l(t) = [M^l(t)]_{1:n} \cdot W_V$;
6        $h_v^{l(j)}(t) = \text{Attn}^{(j)}(Q^l(t), K^l(t), V^l(t))$, $0 \le j < k$;
7        Aggregate multi-head attention output $h_v^{l(j)}(t)$;
8     $h_v^l(t)_{norm} = \dfrac{h_v^l(t)}{\|h_v^l(t)\|_2}, \forall\, v \in V$;

---

## B  ADDITIONAL RESULTS

Table 9 shows the top-5% accuracy of TATKC under different sampling strategies. The bold values indicate the best results. Degree-based sampling achieves the best performance in four datasets.

## C  CASE STUDY

We extract a subgraph from the tgwiktionary dataset and employ an open graph viz platform Gephi[6] to visualize nodes ranked by betweenness centrality (BC) and TKC. Figure 4 presents the node ranked by both BC and TKC in the case study. Figure 4(a) displays the BC node rankings. Figure 4(b) showcases the TKC node rankings. Nodes with higher TKC/BC scores are represented with darker colors and larger sizes. Nodes 1-10 correspond to editors, the remaining nodes represent articles. It is observed that in Figure 4(a), nodes 5, 3, and 4 are the top-3 important nodes ranked by BC. While in Figure 4(b), nodes 19, 17, and 3 are identified as the top-3 important nodes according to TKC. As can be seen, Figure 4 reflects the disparity between these two centrality measures. Nodes boasting high BC scores are representative of wiktionary editors who act as "bridges" linking various vertices within the graph. While nodes with elevated TKC scores signify influential editors or pivotal wiktionary articles that attract substantial attention from a multitude of editors.

## D  ABLATION EXPERIMENT

We perform experiments to dissect various components of TATKC. Multiple variations are created to gain deeper insights into the respective contributions and efficacy of components. We analyze the practical effects of time-injected self-attention and continuous time encoding.

---

[6]Gephi is available at https://gephi.org/

**Table 10: Top-1% and Top-5%(×0.01) ablation results of continuous time encoding**

| datasets | Top-1% | | | Top-5% | | |
|---|---|---|---|---|---|---|
| | Time encoding | Position | Empty | Time encoding | Position | Empty |
| edit-tgwiktionary | **90.68** | 52.50 | 0.53 | **93.77** | 80.14 | 5.29 |
| edit-mlwikiquote | **85.27** | 73.33 | 1.24 | **89.37** | 56.52 | 5.64 |
| edit-mgwikipedia | **80.01** | 73.90 | 0.55 | **86.67** | 60.36 | 8.96 |
| edit-plwikiquote | **85.34** | 79.22 | 4.25 | **85.08** | 80.65 | 7.99 |
| edit-ltwiktionary | **88.14** | 52.80 | 1.42 | **94.32** | 60.21 | 14.1 |
| edit-zhwiktionary | **72.20** | 44.20 | 0.18 | **91.48** | 51.48 | 5.04 |
| edit-warwikipedia | **91.35** | 74.32 | 0.48 | **95.74** | 95.55 | 6.22 |
| edit-mgwiktionary | **90.25** | 38.79 | 0.15 | **89.81** | 64.60 | 4.48 |

**Table 11: Top-1% and Top-5%(×0.01) ablation results of time-injected self-attention mechanism**

| datasets | Top-1% | | | Top-5% | | |
|---|---|---|---|---|---|---|
| | Self-attention | LSTM | Mean | Self-attention | LSTM | Mean |
| edit-tgwiktionary | **90.68** | 85.23 | 86.83 | **93.77** | 92.86 | 90.97 |
| edit-mlwikiquote | **85.27** | 85.17 | 70.99 | **89.37** | 89.28 | 89.02 |
| edit-mgwikipedia | **80.01** | 71.88 | 34.35 | **86.67** | 83.11 | 81.80 |
| edit-plwikiquote | **85.34** | 81.06 | 80.54 | **85.08** | 82.53 | 81.53 |
| edit-ltwiktionary | **88.14** | 86.14 | 57.34 | **94.32** | 93.54 | 93.33 |
| edit-zhwiktionary | **72.20** | 71.04 | 46.83 | **91.48** | 91.11 | 91.46 |
| edit-warwikipedia | **91.35** | 88.43 | 80.33 | **95.74** | 95.26 | 94.50 |
| edit-mgwiktionary | **90.25** | 85.41 | 76.28 | **89.81** | 86.82 | 88.27 |

## D.1 Continuous Time Encoding

To assess the impact of continuous time encoding, we substitute it with a learnable positional embedding, similar to the approach in [40], or simply replace it with a zero vector as empty embedding. The results are presented in Table 10. It is seen that continuous time encoding achieves the best performance across all datasets, both in the top-1% and top-5%. Positional embedding achieves the second-best result, while empty embedding performs the worst in all datasets. This result proves time information is vital to predict node rankings in terms of TKC and continuous time encoding could help self-attention learning time information. Positional encoding can introduce temporal information by sorting neighbors based on timestamps, but it cannot capture the complex time difference information involved in TKC computation. Empty embedding is equivalent to not providing any information about node order or time, resulting in significant information loss, and rendering TATKC incapable of learning the temporal relationships between nodes.

## D.2 Time-injected Self-attention Mechanism

We substitute the proposed time-injected self-attention mechanism with either a mean pooling or LSTM module. Both of these alternatives are commonly employed for sequence encoding. The results are presented in Table 11. First, It is observed that the time-injected self-attention achieves the highest performance across all datasets, both in the top-1% and top-5%. LSTM ranks as the second-best performer in most datasets. For example, in terms of top-1% accuracy, LSTM outperforms Mean pooling in 7 out of the 8 datasets. This indicates that considering both the time sequence and neighbor contributions is beneficial for predicting the top-N% most important nodes. However, though LSTM is capable of capturing time sequence information, it falls short in learning neighbor contributions. Mean pooling, unlike LSTM, cannot learn either time sequence or neighbor contributions, leading to unstable and poor performance.

Received 12 October 2023; revised xx xx 2023; accepted xx xx 2024

