# OpenReview forum: "TATKC: A Temporal Graph Neural Network for Fast Approximate Temporal Katz Centrality Ranking"
_ACM.org/TheWebConf/2024/Conference — TheWebConf24_

### Official Review · Reviewer_qLan · 2023-11-15

**Novelty:** 3
**Technical Quality:** 2

**Review:**

Summary:

This paper introduces TATKC, a novel framework aimed at making predictions of TKC rankings for graphs by better integrating the features of neighbor nodes and improving computational efficiency with a degree-based neighbor sampling method. The framework comprises two main components: Node representation learning, which utilizes continuous time encoding and a time-injected self-attention mechanism to learn node representations, and TKC ranking prediction, which utilizes the learned node representations to predict node rankings based on TKC values with MLP. Extensive experiments show the effectiveness of TATKC.

Strengths:

The studied research problem temporal Katz centrality computation is essential.

Extensive experiments show the effectiveness of the proposed method.

Weakness:

1. Some design choices are rather arbitrary and not well supported by explanations/rationale. For example, why a two-layer MLP is specially selected to be used in the time-injected self-attention mechanism?

2. The paper is not well written and sometimes confused. The writing of the paper needs to be improved to better understand its contributions. For example, Existing temporal neural networks are mostly tailored for specific tasks other than TKC, hence they often lack the capability to effectively and efficiently simulate the TKC computation process.

To accurately simulate the temporal walks, assign weights to them, and accumulate the contribution from all weighted temporal walks, we employ a self-attention mechanism in the modeling process.

It is difficult to understand why existing methods lack the capability to effectively simulate the TKC computation process and why assigning weights to the network can simulate the temporal walks, please explain them more clearly.

**Questions:**

1. By integrating so many up-to-date techniques, the experimental results of the proposed method are slightly better than deep learning models. Which component plays a more important role? Why not compare the proposed method with the traditional methods?

2. There are many hyperparameters in TATKC. It's unclear how sensitive TATKC is to its hyperparameters and how much tuning is required for optimal performance.

3. Why TATKC can fit new data directly without being trained on a new dataset?

4. What is the ratio of training set, validation set, and test set? TATKC is fitted with Ground Truth generated by ETKC, does the process of obtaining the Ground Truth include in the running time?

5. It is recommended to construct a multi-factor fusion metric which has a very high time complexity. It is not recommended to fit traditional algorithms with explicit formulas using graph neural networks.

**Ethics Review Description:**

No issue

**Reviewer Confidence:**

4: The reviewer is certain that the evaluation is correct and very familiar with the relevant literature

**Scope:**

3: The work is somewhat relevant to the Web and to the track, and is of narrow interest to a sub-community

---

### Official Review · Reviewer_mEmr · 2023-11-20

**Novelty:** 5
**Technical Quality:** 3

**Review:**

This paper focuses on measuring centrality in temporal graphs, which is an important problem in temporal graphs. Specifically, this paper aims to address the high complexity of traditional temporal Katz centrality and proposes a method named TATKC that utilizes temporal graph neural networks to compute approximate results. To the best of my knowledge, this paper is the first attempt to approximate temporal Katz centrality using deep learning methods.
## Strengths：
1. This paper leverages temporal GNN to approximate temporal Katz centrality ranking and presents a model named TATKC. TATKC is the first learnable method for approximating temporal Katz centrality, which is novel.
2. The proposed TATKC is introduced in detail and is easy to follow.
## Weaknesses：
1. This paper introduces TATKC to approximate temporal Katz centrality, however, it does not provide an error analysis regarding this approximation. Therefore, I argue that TATKC may not be technically sound.
2. Table 2 shows that L2 normalization has a significant impact on the performance of TATKC, but the paper lacks a corresponding analysis.
3. The experiments in Tables 2-6 do not provide the sampling strategy used.
4. The paper does not compare with baselines such as DRBC and GNN-BET on all datasets, as shown in Tables 4-6, and there is no corresponding explanation provided.
5. Figure 2 could be enlarged to improve visibility.

**Questions:**

See Review.

**Reviewer Confidence:**

3: The reviewer is confident but not certain that the evaluation is correct

**Scope:**

4: The work is relevant to the Web and to the track, and is of broad interest to the community

---

### Official Review · Reviewer_1BsF · 2023-11-22

**Novelty:** 3
**Technical Quality:** 5

**Review:**

The manuscript presented a neural method with self-attention and MLP to fast approximate katz centrality ranking for temporal networks. The problem, fast temporal katz centrality, itself is indeed needed to solve for general applications.  However, the presented methodology itself is not novel enough; and did not solve the centrality computation but just rankings. Basically the encoding part is almost identical to the classic self-attention mechanism with the only exception that instead of encoding positions in sequential data, the presented model encodes time, which essentially is still sequential data.  The final ranking is learned through a simple MLP model. The manuscript is very well written and easy to follow. The analysis of results is thorough and detailed.

**Questions:**

- Fig 1. I may have missed it. But how is the value of L layers decided?
- Line 558, based on Eq. (6) isn’t the space complexity a function of time steps t_max? Especially when t_max is much larger than the sample size, which can be true for many application cases.
- Line 613, TGAT is not tuned for node ranking, but mostly for node representation. So it is not surprising that the presented model has better performance, shown in Table 2.

**Reviewer Confidence:**

4: The reviewer is certain that the evaluation is correct and very familiar with the relevant literature

**Scope:**

3: The work is somewhat relevant to the Web and to the track, and is of narrow interest to a sub-community

---

### Official Review · Reviewer_Dy3B · 2023-11-23

**Novelty:** 5
**Technical Quality:** 4

**Review:**

**Short summary**

The paper addresses the problem of computing the ranking of nodes in a temporal network according to the Temporal Katz-Centrality (TKC) scores. TKC of a node, by definition, is computed over the temporal walks of a temporal network, considering the timing over the edges of such walks. The authors propose an architecture to obtain representations of nodes that consider the timings and structure of the networks (i.e., capturing the temporal walks that go through a specific node). Then they use such learned representations to rank the nodes according to their predicted TKC. Experiments are performed to show the efficacy of the proposed architecture in predicting node rankings according to TKC, and experimental evaluation of various model design parameters.

**Strengths**
1. The problem addressed in this work is innovative and not much work has been done in the area.
2. The proposed ranking algorithm seem to be useful in practice compared with existing baselines employed by the authors.
3. The proposed method seem to be quite scalable and attain good generalization.

**Weaknesses**
1. Obtaining temporal Katz-centrality scores is far from being established as an important primitive for problems regarding temporal networks.
- While temporal centrality measures are for sure important for temporal network mining, it is currently far from established to which extent TKC scores can be useful in practice and their impact in web scenarios. The authors should better explain some specific applications of how the rankings provided by their model can be used in practice.
2. To my understanding only rankings of the various nodes according to the TKC can be predicted using the model proposed by the authors.
Therefore, it is not clear to me if the proposed approach can estimate with high accuracy the temporal Katz-centrality values of the different nodes. This can be an important thing to specify, in particular if this is the case, then metrics reporting the approximation errors (such as average and absolute errors) should be describe in the experimental part (currently only rankings are evaluated, e.g., Kendall’s tau). If instead the method cannot be used to obtain the values of TKC for the various nodes, then this is a strong limitation of the proposed method since in practice one usually wants to obtain a pair $(v, C(v))$ for each node $v$ where $C(v)$ is an estimate of the TKC of node $v$.

3. Experimental part.
- Authors should consider more appropriate baselines:
The baselines adopted by the authors to which they compare their architecture are not specifically designed for the proposed problem. Mainly, such baselines are developed for betweenness centrality (not even temporal betweenness centrality under optimal walks), so it is not surprising that the proposed approach outperforms all such models. The authors can address this issue by, for example, devising some simpler architecture for the problem at hand, and also consider some simple combinatorial algorithms without guarantees. A combinatorial algorithm can consider, for example, only temporal walks up to a certain size, or only optimal shortest temporal walks.
- More diverse datasets are needed.
The authors consider only dataset from one main domain when evaluating their proposed architecture, this does not guarantee that the proposed approach generalizes well on many different domains. To this end, I would suggest to the authors to consider many possible datasets from other domain, e.g., social networks, phone call networks etc (that are easily available online).

4. Some reproducibility issues arise.
The authors do not fully discuss the details of their implementation (e.g., the exact datasets they used to train their architecture) and the code is not available for review, and the authors do not mention to publish it upon acceptance.

**Minors**
- Figure 1 should be described properly.
- Line 399, $n$ is used for the size of the neighbourhood, while previously it was used for the walk length
- Use \eqref across the manuscript to reference formulas.
- line 405 all projection matrices should be described properly
- line 535 the values $y_i, i=1,2$ are not clear to me since they were never mentioned previously.

**Questions:**

If the authors can comment on the weak points 2,3, and 4.

**Ethics Review Description:**

No ethical issues

**Reviewer Confidence:**

3: The reviewer is confident but not certain that the evaluation is correct

**Scope:**

3: The work is somewhat relevant to the Web and to the track, and is of narrow interest to a sub-community

---

### Official Review · Reviewer_iQAw · 2023-11-24

**Novelty:** 6
**Technical Quality:** 6

**Review:**

The authors of this work use a temporal graph neural network to predict a ranking of nodes that is based on a temporal generalization of Katz centrality for time-stamped network data. The proposed model utilizes a self-attention mechanism, in which the common positional encoding is replaced by a temporal encoding that considers the sequences of nodes in temporal walks. By this, different from other works that have used temporal graph neural networks, the proposed approach is able to capture patterns that unfold in the temporal ordering of nodes, rather than being based on a sequence of time-aggregated snapshots that discard information in the temporal ordering. The model is further based on an inductive approach that allows to learn on small temporal graphs and generalize to massive-scale graphs.

This is a well-written paper with promising results that makes a strong contribution to the field. The model and its underlying motivation is explained well and the authors consider the time and space complexity of both the training and prediction phase, highlighting that their approach provides a six-fold speed-up over the exact calculation of temporal Katz centrality. The experimental setup is described well, the data sets and I consider the reproducibility of this work as high. The application scenario and the data sets are clearly relevant for the WWW community.

In summary, I believe that this work is a worthwhile contribution to the Web Conference, but I would like to ask the authors to consider my questions below.

**Questions:**

Please list questions for the authors for the discussion period, involving issues that an author response could change your opinion, clarify a confusion, or address a limitation.

1) The definition of temporal walks in section 3 only considers the temporal ordering of edges and not their temporal distance. This can actually limit the information that we obtain from a temporal network, as walks across two interactions that occur within seconds are considered equal to walks across two edges that lie days or weeks apart. Moreover, neglecting time stamps in the walk definition can lead to an effect that basically correspond to a time aggregation insofar as *any* future interaction can continue walks, while in reality paths relevant for a given process (e.g. rumor spreading) are more constrained. To address this, it is common to consider causal or time-respecting paths with a limited waiting time, which effectively fixes the time scale of the process that we consider. A thorough discussion of this issue can be found in the temporal network analysis literature.

It would be good to add this discussion to the paper, highlighting that the inclusion of limited waiting times could considerably change the results.

2) To the best of my knowledge, I agree with the authors' assessment that this work is the first to consider a learning-based prediction of temporal Katz centrality, which highlights the novelty and timeliness of the contribution. However, I do think there are a couple of related works that have used (learning-based) approaches to approximate other (temporal) centrality measures, which are not mentioned, e.g.

[1] Riondato, M. & Upfal, E. Abra: Approximating betweenness centrality in static and dynamic graphs with rademacher averages. ACM Transactions on Knowledge Discovery from Data (TKDD)
[2] Bergamini, E., Meyerhenke, H. & Staudt, C. L. Approximating betweenness centrality in large evolving networks. In 2015 Proceedings of the Seventeenth Workshop on Algorithm Engineering and Experiments (ALENEX), 133–146 (SIAM, 2014).
[3] Grando, F. & Lamb, L. C. Estimating complex networks centrality via neural networks and machine learning. In 2015 International Joint Conference on Neural Networks, IJCNN 2015, Killarney, Ireland, July 12-17, 2015, 1–8 (IEEE, 2015)
[4] Grando, F., Granville, L. Z. & Lamb, L. C. Machine learning in network centrality measures: Tutorial and outlook. ACM Comput. Surv. 51 (2018).
[5] Santoro, D. & Sarpe, I. Onbra: Rigorous estimation of the temporal betweenness centrality in temporal networks. In Proceedings of the ACM Web Conference 2022

Moreover, closest to the approach and motivation considered in this paper, there is a very recent work on arXiv that has used a temporal graph neural network to predict temporal betweenness and closeness centralities:

[6] Heeg, F. and Scholtes, I.: Using Causality-Aware Graph Neural Networks to Predict Temporal Centralities in Dynamic Graphs, https://arxiv.org/abs/2310.15865

Given that [6] was uploaded to arXiv after the submission deadline, I obviously do not consider this an omission of related work, but given that it is very closely related it nevertheless contributes to the motivation of the authors' work, highlighting its relevance and timeliness.

3) A third and final question is about the baseline methods. I agree that there is no direct competitor in GNN-based temporal Katz centrality prediction that can serve as a natural baseline and I specifically appreciated that the authors chose to compare their approach to multiple temporal graph neural network architectures, some of which can be considered as ablation studies of the proposed architecture. I just have two remarks:

- the discussion of the specific choice of baseline methods, and what we can learn from their different performances compared to the proposed self-attention model, could be improved
- it would be interesting to see the performance compared to a *static* GNN architecture, as this would highlight the relevance of the temporal patterns compared to the mere static topology of links.
- related to the previous point, I would have been interested to see the difference of (ground truth) temporal Katz centrality rankings compared to the temporal generalization.

Both of those points are actually related to question 1, as it is unclear to me whether the implcit aggregation that is due to the absence of a limited waiting time for walks could lead to the fact that temporal walks are in deed very similar to walks in the static topology.

In summary, I am in favor of this work and would be willing to further increase my score if the questions above are addressed.

**Reviewer Confidence:**

4: The reviewer is certain that the evaluation is correct and very familiar with the relevant literature

**Scope:**

4: The work is relevant to the Web and to the track, and is of broad interest to the community

---

### Decision · Program_Chairs · 2024-01-22

**Decision:**

Accept

**Comment:**

There was quite a bit of discussion on this paper. Overall, I think the merits outweight the downsides, so I vote for a weak accept.

 My main concern (which the reviewers share) is that the Temporal Katz Centrality Ranking is not justified or motivated enough. The paper should make a good case about why this is important. Also, the method only gets the ranking, not the scores. So, the motivation section should explain why the ranking suffices for applications.

 Reviewer iQAw points out numerous suggestion, which would be good to incorporate in the camera-ready version.